# Umbrella Sampling Simulations of Carbon Nanoparticles Crossing Immiscible Solvents

**DOI:** 10.3390/molecules27030956

**Published:** 2022-01-31

**Authors:** Anastasios Gotzias

**Affiliations:** National Centre for Scientific Research “Demokritos”, Institute of Nanoscience and Nanotechnology INN, 15310 Athens, Greece; a.gotzias@inn.demokritos.gr; Tel.: +30-210-650-3408

**Keywords:** free energy, molecular dynamics, graphene, carbon nanocones

## Abstract

We use molecular dynamics to compute the free energy of carbon nanoparticles crossing a hydrophobic–hydrophilic interface. The simulations are performed on a biphasic system consisting of immiscible solvents (i.e., cyclohexane and water). We solvate a carbon nanoparticle into the cyclohexane layer and use a pull force to drive the nanoparticle into water, passing over the interface. Next, we accumulate a series of umbrella sampling simulations along the path of the nanoparticle and compute the solvation free energy with respect to the two solvents. We apply the method on three carbon nanoparticles (i.e., a carbon nanocone, a nanotube, and a graphene nanosheet). In addition, we record the water-accessible surface area of the nanoparticles during the umbrella simulations. Although we detect complete wetting of the external surface of the nanoparticles, the internal surface of the nanotube becomes partially wet, whereas that of the nanocone remains dry. This is due to the nanoconfinement of the particular nanoparticles, which shields the hydrophobic interactions encountered inside the pores. We show that cyclohexane molecules remain attached on the concave surface of the nanotube or the nanocone without being disturbed by the water molecules entering the cavity.

## 1. Introduction

Many of the pioneering simulations of surface phenomena involve studies of gases and liquids interacting with carbon solids [1,2,3,4]. Gas–solid or liquid–solid adsorption simulations allow assessing information about the micropore size, the surface area, and some aspects of the surface chemistry of sorbent materials [5,6,7,8]. Porous texture characterization based on gas adsorption is associated with a number of classical and advanced simulation tools performing at the microscopic level such as the density functional theory and Monte Carlo simulations [9,10,11]. On the other hand, the characterization of carbons with the help of liquids is especially interesting when the material is intented to be used in a liquid medium. Therefore, it makes sense to use liquids to characterize carbons when they are to be used for instance in liquid purification, liquid–solid heterogeneous catalysis, or liquid suspensions (and not to rely only, despite its popularity for the characterization by gas adsorption) [12,13].

When a carbon nanoparticle enters a solvent environment, the solvent molecules wash the surface and penetrate the accessible cavities. The molecules create a first layer covering the surface with either special composition and/or special interaction with the atomic sites and surface functional groups. Different solvents and solvent blends can be used upon treatment, for instance to deposit solute dopants on the surface of the nanoparticles [14]. The solvation energy is balanced by the fluid–wall interactions for fluids whose surface energy matches that of the walls [15,16]. Hence, hydrophobic carbons are more likely to dissolve in a hydrophobic environment than in a polar one.

Molecular dynamics can offer comprehensive microscopic insight to the solvation of particles by providing statistically sound views of interfacial structure and interaction potentials between solvents and solutes [17]. Most material frameworks bare different chemical functions and may adopt a large number of conformations at the interface. Accordingly, a good knowledge of the surface chemistry and accessibility of the pore network are crucial to realize the affinities that contribute to the solvation process in different environments. It is with this motivation that we simulate the free energy of different carbon frameworks crossing hydrophobic–hydrophilic solvents [18,19]. Practicing with well-defined force fields (all–carbon solids, cyclohexane as the organic solvent, and SPC water) helps the formulation and execution of a sustainable workflow commited to free energy simulations [20,21,22,23,24,25]. The workflow is a precursor on an ongoing computational campaign investigating the binding affinities of such materials embedded in conjugated solvents with larger molecules such as polymers, surfactants, or lipids [26,27].

Due to its ability to assume different hybridization states, including sp, sp${}^{2}$ and sp${}^{3}$ hybridization, carbon may form a miscellany of carbon allotropes with diverse physicochemical characteristics and morphology [28,29]. If one excludes some specific cases such as diamond–like amorphous carbon, these allotropes are mainly constituted of covalently bonded sp${}^{2}$ carbons arranged in hexagonal atomic lattices. Typical carbon frameworks with exclusively sp${}^{2}$ hybridization are the fullerenes, the carbon nanotubes, and the graphene. These nanoparticles can be configured in single–layer or few–layer formations and therefore obtain a high specific surface area, which is essential for energy-storage applications. Due to their well-defined geometry, these nanoparticles can be used as starting materials for designing different nanoscale low-dimensional solids by controlling their dimensionality size and shape [30]. Given also their excelent electrical, mechanical, and thermal properties, they are regarded as emerging materials for sensors, semiconducting devices, and transparent electrodes. It is also impressive that these nanoparticles can potentially be exploited in high-priority biorelated fields. This has recently motivated research on the biocompatibility of the carbon interface and investigation of how carbons may replicate the affinity and transport properties of proteins [31,32,33,34,35,36,37].

A particular class of sp${}^{2}$ hybridized carbon nanoparticles that has received little attention is that of carbon nanocones (CNCs) [38,39,40,41]. CNCs were synthesised by accident in 1995 in the so-called Kværner Carbon Black and $H_{2}$ process, which decomposes hydrocarbons directly into carbon and $H_{2}$ [42]. Nowadays, conical carbon nanoparticles are produced through the pyrolysis of hydrocarbons in a plasma torch process [43]. CNCs moderately resemble open half fullerenes. The framework of the cones consists of regular hexagonal carbon rings and one to five pentagonal rings concentrated near the tip [44,45,46]. The cones occur in five different apex angles, depending on the number of the pentagons on the framework. Carbon nanocones as well as nanohorns (i.e., nanotubes capped with sharp cones at one end) have been proposed for a number of applications such as gas storage and heterogeneous catalysis and have been suggested as probes in electronic and optical nanoscale devices [47,48,49,50,51,52].

We build molecular models of three carbon nanoparticles, namely a carbon nanotube, a graphene sheet, and a carbon nanocone. We configure a biphasic system consisting of cyclohexane and water. We embed the carbon nanoparticle into the cyclohexane layer and push it to cross the interface and enter the water layer. We perform umbrella sampling simulations along the charted path of the nanoparticle and compute the solvation energy difference between the two solvents [53,54]. The water-accessible surface area of the nanoparticles along the path is also discussed. We demonstrate that cyclohexane remains adsorbed inside the confinement, protecting the hydrophobic carbon surface on the concave side from the more unfavorable interactions with the polar water molecules.

## 2. Materials and Methods

To generate the coordinate files of the graphene and the nanotube, we used the BuildCstruct python script [55,56]. We built a square graphene sheet with edge α = 3.0 nm and an armchair nanotube, CNT(10,10) having a radius, r = 0.68 nm and height, h = 4.5 nm. We built the carbon nanocone by removing n${}_{p}$ = 3 sectors of 60${}^{{}^{\circ}}$ from a flat graphene disc and connecting the dangling bonds. For each removed sector, we introduced one pentagon at the tip. The pentagons were topologically isolated, meaning that each pentagonal face was surrounded by five hexagons. This gave an appropriate surface curvature near the apex of the cone. The cone angle, θ, is given by the expression, $sin\left(\theta /2\right)=1-n_{p}/6$. Setting n${}_{p}$ = 3, we obtained θ = 60${}^{{}^{\circ}}$. The base radius of the cone was r = 2.09 nm. Due to the apex curvature, the effective height of the cone was h${}_{c}=3.55$ nm, that is, slightly smaller than the actual geometric height of the cone h${}_{c}$ = r/tan(θ/2)=3.63 nm. The modeling details of the carbon nanotube, the graphene, and the nanocone are listed in Table 1.

We modeled the $sp^{2}$ carbon atoms on the basis of the OPLSAA reference of naphthalene and of aliphatic carbons. Hydrogen atoms were modeled as benzene hydrogens. The carbon structures were uncharged. We used the GROMACS package, version 2020 to carry out the simulations. The simulations were submitted to the high-performance computing services of the Greek National Infrastructure for research and technology, GRNET-ARIS.

We placed the carbon nanoparticle at the center of a cubic simulation box with size *l* = 8 nm. Selecting this size, we ensured that the nearest edge of the cube was distant more than 1.5 nm from the outermost carbon atom of the nanoparticle. The cut-off radius of the pair interactions was set to 1.4 nm. We used the *insert* command of GROMACS to insert 5500 cyclohexane molecules randomly into the *l*-cubic box. In all cases, less than 4500 molecules were successfully inserted. The CH${}_{2}$ nodes of the cyclohexane molecules were modeled as alkane carbons. We applied position restraints on the carbon nanoparticle and let the cyclohexane molecules relax using a steepest descent minimization followed by two equilibration steps, first in the NVT and then in the NPT ensemble over 0.2 and 1 ns, respectively. The pressure was regulated at 1 bar and the temperature was regulated at 310 K. Due to the pressure coupling, we expected the box to isotropically expand during the NPT equilibration, so that at the end of the simulations, the cube edge, *l*, became larger than 8 nm.

To build the biphasic system, we edited the box sizes to (X × Y × Z) = (*l* × *l* × 5/2
*l*), where *l* is the cube size after the last NPT equilibration. In the output, the bottom *l*-cubic section of the box contained the equilibrated cyclohexane layer with the carbon nanoparticle at the center, while the upper *l*×*l*×3/2
*l* section of the box was empty. The empty rectangular section was solvated with SPC water. In order to avoid placing water molecules into the cyclohexane layer, before triggering the *solvate* command of GROMACS, we changed the carbon (C) radius from 0.17 to 0.35 nm. For the rest of the simulation, the C radius was reset to 0.17 nm. As previously, the molecules of both solvents were relaxed, first using a steepest descent minimization, then an NVT equilibration over 0.5 ns and next an NPT over 1 ns. The pressure and the temperature were set to 1 bar and 310 K, respectively. After completing the equilibration steps, the box sizes and the number of the solvent molecules described therein are listed in Table 2.

To generate the initial configurations of the pertinent umbrella sampling simulations, we performed a pulling simulation. The pulling simulation continued from the last NPT equilibration. We removed the position restraints and pulled the carbon nanostucture along the z coordinate, using a force constant k = 1000 KJ mol${}^{1}$ nm${}^{-2}$. To specify the pulling direction, we drew a line connecting the COM of the carbon structure and a CH${}_{2}$ node of a random cyclohexane molecule located above the structure (i.e., close to the interface, near the center of the box). Then, the pulling direction was given by the z-projection of the line. We used a pull rate 10 nm ns${}^{-1}$ and let the pulling simulation run over 1.5 ns. The carbon structure, the water molecules, and cyclohexane were indexed in different groups. The groups were coupled in separate temperature coupling paths. We applied the Nose–Hoover method for the temperature coupling and the Berendsen barostat for the pressure. The reference temperature and pressure were 310 K and 1 bar, respectively. We set the period of the temperature and pressure fluctuations to 1 picosecond. The timestep of the simulations was 1 fs.

The pulling simulation indicated a path through which the carbon nanoparticle traveled a distance, 10 nm (on the z axis), passing over the interface of cyclohexane and water. We used a default step size, dz = 0.23 nm, to collect a number of configurations along the path. We collected additional configurations at the region of distances where the carbons laid close to the interface, using smaller step sizes such as dz = 0.11 nm and dz = 0.06 nm. The size and the number of the smaller steps depended on the width of the nanoparticle and the time it required to cross the interface. The selected configurations were given as inputs to the umbrella sampling simulations. The umbrella simulations started by running a brief, 0.2 ns, NPT equilibration step. The equilibrated configurations were passed into the long umbrella simulations (NPT), which ran over 10 ns. We used the Parrinello–Rahman pressure coupling to generate a rigorous NPT ensemble. Depending on the carbon nanoparticle, we performed a total of 45 to 60 umbrella simulations and generated a histogram of energy distributions (umbrella windows) throughout the charted path. We verified that the energy distributions were overlapping between any pair of consecutive (z, z + dz) positions along the path. The umbrella simulations were submitted as an array of jobs and ran simultaneously on the computing nodes of GRNET. The computing performances of the umbrella simulations are listed in Table 3. After completing the simulations, we calculated the potential mean force (PMF) of the nanoparticles throughout the path, following the weighted histogram analysis method (WHAM) [57].

An interesting variable that can be continuously recorded during the course of the umbrella sampling simulations is the solvent-accessible surface area of the nanoparticles. This variable depicts how much of the surface is exposed to a particular solvent. We computed the solvent-accessible surface area using the double cubic lattice method of Eisenhaber et al. [58]. The van der Waals radii of the atoms were taken from Bondi [59]. We grouped the non-water components of the system (i.e., cyclohexane molecules and the carbon particle) and set the carbon particle as the solute, for which we computed the surface area that was in contact with the different solvent (i.e., water).

## 3. Results

In Figure 1, we present a sequence of configurations from the pulling simulations performed on three carbon nanoparticles: namely, a nanotube, a graphene, and a nanocone. The box is solvated by two-fifths with cyclohexane and three-fifths with water. We apply a force to pull the carbon samples on the z direction. At z = 0, the samples lay at the center of the cyclohexane layer. The nanotube preserves an almost horizontal orientation up to the moment it crosses the interface. Within the water environment, the nanotube tends to align with the pulling coordinate, letting the cyclohexane molecules entrained by it drop back into the cyclohexane layer. Likewise, the graphene is initially planar to the interface. After crossing the interface, the graphene rotates nearly at a normal direction to align with the pulling coordinate. On the other hand, the nanocone obtains a clockwise motion as it moves on the z direction. When the nanocone enters the water environment, the tail shifts down, dropping some of the cyclohexane molecules carried inside the cavity. At the final configuration (z = 10.9 nm), the carbon samples reach aproximately a distance 5 nm from the interface. At this distance, the samples are totally dissolved in water as they do not interact with the cyclohexane molecules at the bottom layer. The interface of the solvents appears flat, while some of the stray cyclohexane molecules previously found in the water section have returned to the layer from which they originated. In all configurations, there is an amount of cyclohexane molecules located at the top of the water layer. This is an effect of the periodic boundary conditions applied in the box.

Figure 2 shows the force applied to pull the carbon nanotube, the graphene, and the nanocone from the cycloexane layer into water. The force is expressed as a function of the center-of-mass (COM) position, z, of the carbon nanoparticle due to the pulling. The pull rate is 10 nm ns${}^{-1}$. The force is initially small because the particles can aimlessly move within the hydrophobic (cyclohexane) environment at the specified pull rate. The pull force increases when the carbon particles approach the interface where the water–carbon interactions start to develop. When the carbons pass over the interface, the nearby bulk molecules are disturbed, causing the solvent–solvent interactions between the two solvents to enhance. As the particles move further on the z direction, they rotate in an appropriate direction to reduce the amount of hydrophobic interactions at their front. The particular orientation makes the force decrease.

The advantage of the pulling simulations is that it allows creating a path of interest in a single, rapid simulation. For instance, a pulling simulation over only one nanosecond is sufficient to make the carbon nanoparticles travel a distance of 10 nm, penetrating the interface of cyclohexane and water. This is accomplished by setting an adaptive force and a pull rate (10 nm ns${}^{-1}$) to help the nanoparticles overcome a free energy barrier and cross the interface. If it were not for the pull force, the interface would never be crossed, or it would be crossed with an appreciable hysteresis. Therefore, pulling simulations prepare the system for the presence of an interface, charting a path through which the carbon nanoparticles may pass. This path is statistically insignificant, since the particles can likely reach the same final state through infinite possible paths. In order to explore the relevant configuration space with statistical efficiency, we perform umbrella sampling simulations. Configurations such as those shown in Figure 1, picturing the position of the nanoparticle across the pulling path, become inputs to a sequence of individual umbrella simulations.

A selective set of umbrella sampling simulations performed on the carbon nanotube, the graphene, and the nanocone is shown in Figure 3, Figure 4 and Figure 5, respectively. The carbon nanoparticles lay at certain positions, z, near the interface. In order to realize the configuration space of the umbrella sampling, we depict the carbon nanoparticles in 10 timestep configurations during the runs. In all representations, the timestep size is 1 ns and the simulations run over 10 ns. The COM of the carbon nanoparticles is centered in the box. We use a red–blue color palette to display the nanoparticles at the specified timesteps. The timestep configurations resemble those of the pulling simulation, as detailed in Figure 1. That is, in the cyclohexane layer, the carbon particles lay relatively planar to the interface, whereas inside the water environment, they rotate at a nearly normal direction to align with the z axis. Based on this outcome, we infer that the initial configurations, created by the pulling simulation, are indeed, the most frequently sampled configurations in the umbrella simulations. This confirms that the systems are well equilibrated during the NPT simulations, so that umbrella sampling is restrained in narrow energy distributions (umbrella windows), producing qualitatively similar configurations.

The panels on the second row in Figure 3, Figure 4 and Figure 5 show the water-accessible surface area (WASA) of the carbon nanoparticles recorded during the same umbrella simulations as in the illustrations on the first row. WASA values present strong fluctuations with time, because the cyclohexane molecules are randomly exchanged with water on the fluid–wall interface during sampling. WASA is recorded for all the performed umbrella simulations. The time average of the WASA recordings expressed as a function of the COM position z of the carbon particles is shown in Figure 6. The nanoparticles are initially out of the range of water interactions, and therefore, their surface is inaccessible to water. When the carbons cross the interface, the cyclohexane molecules sweep gradually from the walls, leaving parts on the surface water accessible. Regarding the example of the graphene sheet, the top side of the sheet is wetted first, and the other side is wetted only after the graphene enters adequately the water layer and the sheet begins to rotate. Inside the water layer, the rotation of the carbon particles facilitates the fast rejection of the cyclohexane molecules from the walls. Presumably, when the particles are more aligned to the z axis, the hydrophobic interactions are better screened on both sides of the walls.

In Figure 6, WASA values are compared to the geometrical surface of the nanoparticles, as listed in Table 1. In the case of the graphene, we detect a complete wetting of the surface near the final state. There, the water-accessible surface area becomes twice the 3 nm square area of the sheet. This confirms that our calculations are consistent, because WASA is recorded using the same van der Waals radii in the atoms definition with the NPT simulations. On the other hand, WASA recordings of the nanotube show that a small amount of cyclohexane molecules is never rejected from the surface. Although the external surface of the nanotube ($S^{out}$) becomes fully wet, cyclohexane molecules form a strict hydrophobic structure inside the cavity ($S^{in}$), which is hardly disturbed by the water molecules entering the nanopore. Accordingly, WASA recordings of the nanocone show that this structure wets only the external surface. When the nanocone enters the aqueous environment, it shifts the conic tip upward, shielding the cyclohexane molecules behind the wall. Since cyclohexane is more attracted than water by the carbon sites, it creates a monolayer coverage over the concave wall, making the internal surface of the cone inaccessible to water. Notably, the water molecules restrict the cyclohexane structure to grow up to the thickness of a monolayer. We do not expect the cyclohexane to condense inside the conic pore, although it should be more concentrated near the apex, where the confinement is narrower.

In Figure 7, we present the potential mean force (PMF) of the carbon nanoparticles as a function of the COM position within the box. The plateau on the curves indicates that the particles are completely dissolved in water. This is achieved when the particles reach a distance larger than the cut–off from the cyclohexane–water interface. Lengthier and wider nanoparticles need to move further inside the water layer to dissolve. This explains why we set a larger height for the water layer than the cyclohexane when editing the box sizes. As the structures move, the nearby solvent molecules agitate. The agitations change the entropy and contribute to the system energy. The nanotube and the nanocone need considerably higher energy than the graphene to dissolve. It is because these structures create greater agitations to the ambient molecules than the graphene. It is also because the cone and the nanotube contain a nanopore. Cyclohexane molecules remain attached on the walls of the nanopore, even when these structures enter suifficiently the water layer. The adsorbed cyclohexane monolayer provides extra hydrophobic interactions in the water layer along with those counted on the interface of the solvents. The excess of the hydrophobic interactions increases the potential energy.

## 4. Discussion

Realizing the properties of carbon sorbents in a solvated chemical environment requires first investigating the interfacial structures and interaction potentials with solvents having contradictory characteristics. For instance, cyclohexane and water differ in hydrophilicity and polarity. When a porous hydrophobic carbon is disperced in cyclohexane, the cyclohexane molecules remain adsorbed inside the nanopores even after the sorbent is pushed into the aqueous layer. Inside the nanoconfinement, the hydrophobic molecules assume the role of pseudosurfactants, shielding the carbon wall from the interactions with the polar water molecules. This outcome, realized by umbrella sampling molecular dynamics, suggests an interesting direction for experimentation and may eventually help in the interpretation of the relevant results. Taking advantage of the simplicity of the carbon surface, we establish a lower bound on the sampling necessary to simulate more complex low-dimesional materials such as functionalized graphenes, metal oxides, and metal–organic nanosheets (MONs). This work is a precursor of an ongoing large simulation study of the binding affinities of these solids in environments with conjugated solvent molecules such as surfactants, ionic liquids, and lipid bilayers.

## 5. Conclusions

We performed umbrella sampling molecular dynamics to simulate three carbon nanoparticles with different shapes crossing the interface of cyclohexane and water. We used a pull force to direct the nanoparticles from the cyclohexane layer to water with the scope to create the initial configurations for the umbrella simulations. Inside the water layer, the nanoparticles aligned with the pulling direction. The particular orientation facilitated the rejection of the cyclohexane molecules from the external walls of the nanoparticles. On the contrary, the surface inside nanoconfinements remained dry. This was because water circumvented the hydrophobic interactions inside the nanopores. We demonstrated that a carbon nanotube and a nanocone needed higher energy than a graphene nanosheet to cross the cyclohexane–water interface. This was attributed to the agitations of the solvent phases due to the movement of the large nanoparticles. It was also attributed to the nanoconfined cyclohexane molecules, which remained adsorbed inside the water layer and contributed to the potential energy.

## Figures and Tables

**Figure 1 molecules-27-00956-f001:**
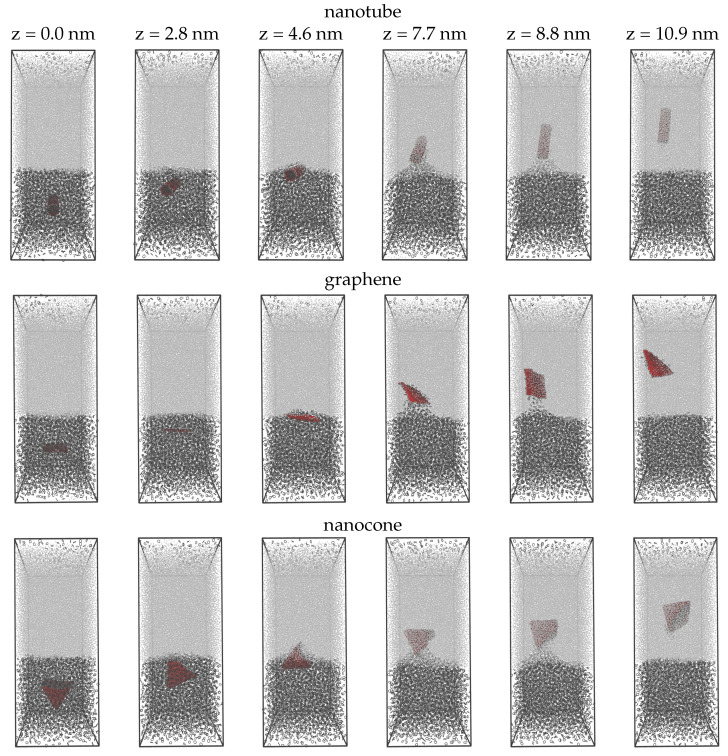
Selected configurations from the pulling simulations of a carbon nanotube, a graphene, and a carbon nanocone, as they move from the cyclohexane layer into water. The snapshots correspond to the different z positions of the nanoparticles. The carbon atoms of the nanoparticles are shown with red spheres. Cyclohexane and water molecules are shown with black and light gray lines, respectively.

**Figure 2 molecules-27-00956-f002:**
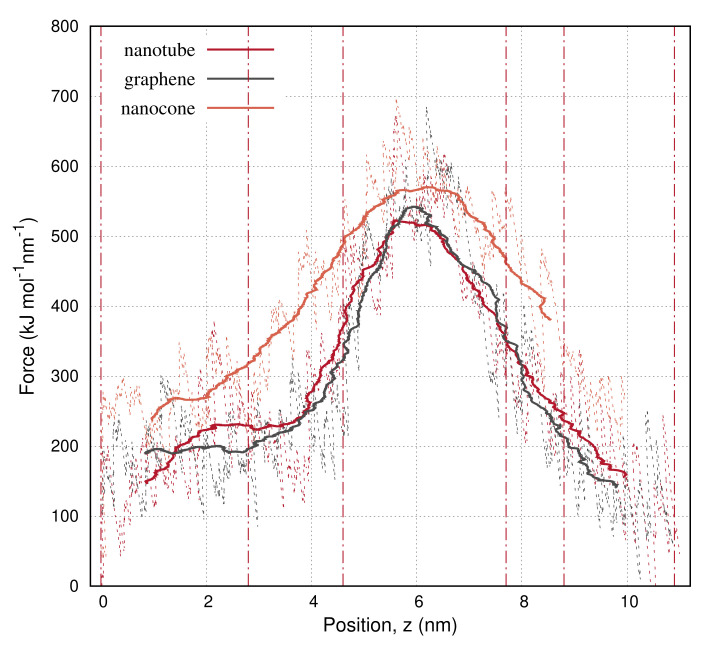
Pull force applied to carry the nanotube, the graphene, and the nanocone, from the cyclohexane layer into water, as a function of the position, z, of the nanoparticle. Lines show the moving average of the applied force over 2000 configurations. The vertical dashed–dot lines depict the positions of the nanoparticles in the configurations shown in Figure 1.

**Figure 3 molecules-27-00956-f003:**
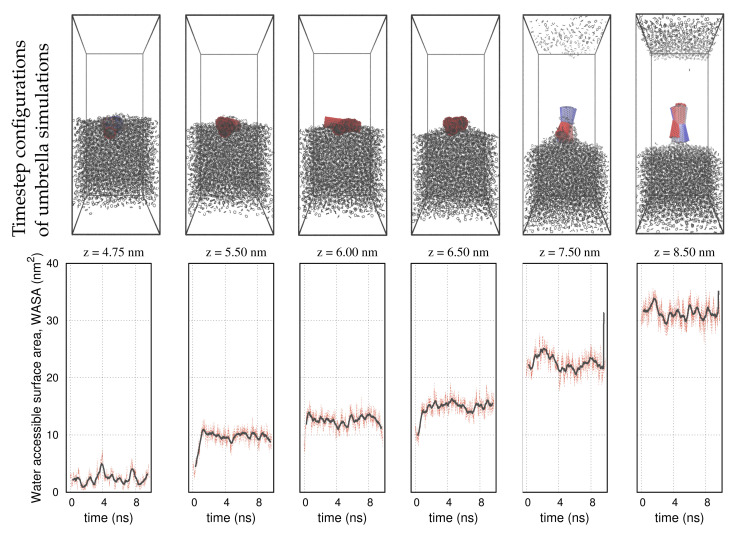
**Up:** Umbrella sampling configurations of the nanotube at different positions and timesteps near the interface of the solvents. The COM of the nanotube is set at the center of the box. The panels present 10 timestep configurations from each simulation. The nanotube is shown with a red–blue color palette. The colors correspond to the different timesteps. Cyclohexane molecules are shown with black lines. Water molecules are not shown for clarity. **Down:** Water–accessible surface area (WASA) of the nanotube during the umbrella sampling simulations. Black lines represent the moving averages of the WASA recordings over 100 sampled configurations.

**Figure 4 molecules-27-00956-f004:**
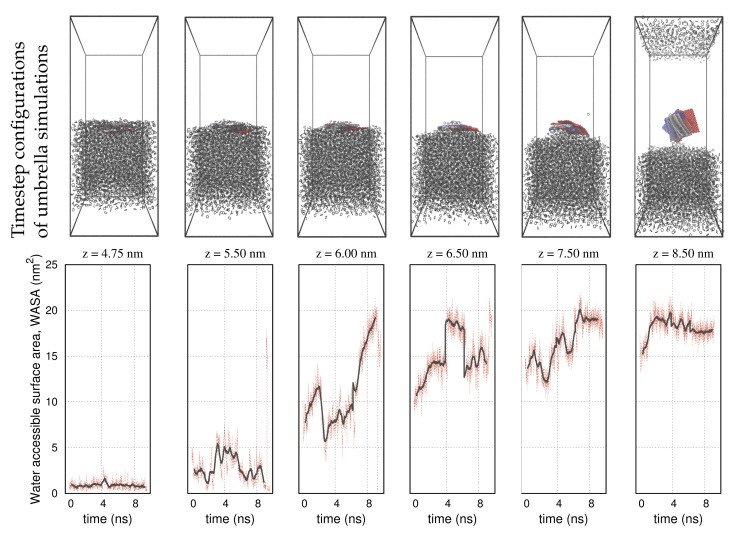
**Up:** Umbrella sampling configurations of the graphene at different positions and timesteps near the interface of the solvents. **Down:** Water–accessible surface area (WASA) of the graphene during the umbrella sampling simulations. We use the same color definitions as in Figure 3.

**Figure 5 molecules-27-00956-f005:**
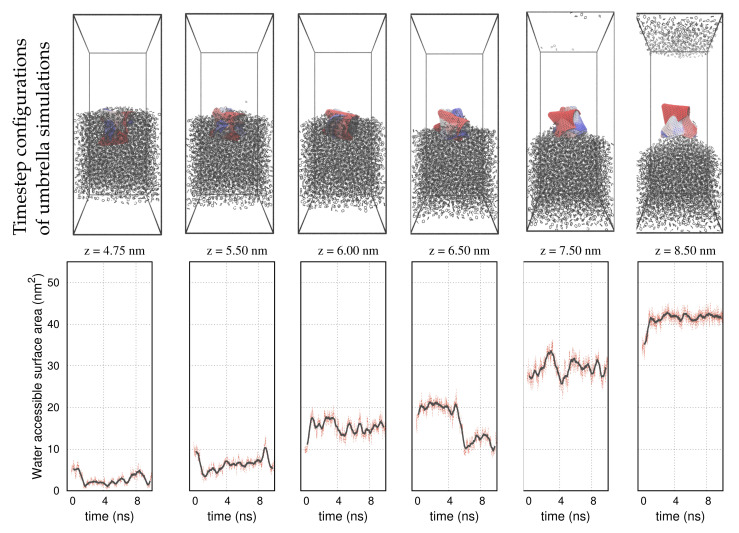
**Up:** Umbrella sampling configurations of the nanocone at different positions and timesteps near the interface of the solvents. **Down:** Water–accessible surface area (WASA) of the nanocone during the umbrella sampling simulations. We use the same color definitions as in Figure 3.

**Figure 6 molecules-27-00956-f006:**
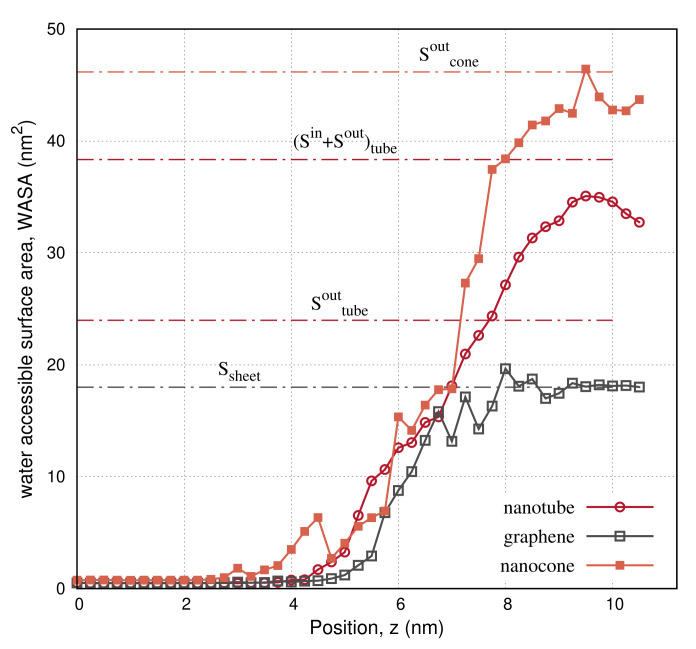
Water-accessible surface area (WASA) of the nanotube, the graphene, and the nanocone, as the nanoparticles move from the cyclohexane layer into water. The dashed–dot horizontal lines depict the external, $S^{out}$, and internal, $S^{in}$, geometric surface of the nanoparticles, as listed in Table 1.

**Figure 7 molecules-27-00956-f007:**
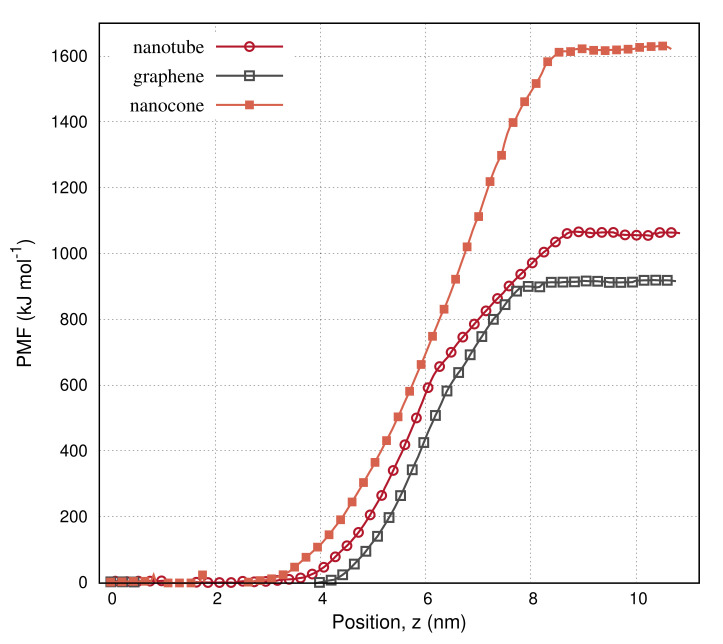
Potential of mean force (PMF) of the nanotube, the graphene, and the nanocone, as the nanoparticles move from the cyclohexane layer into water.

**Table 1 molecules-27-00956-t001:** Modeling parameters of the carbon nanotube, the graphene, and the nanocone and their external, $S^{out}$, and internal, $S^{in}$, surface areas. We used $\sigma_{C}=$ 0.17 nm in the geometrical surface area calculations.

Structure	Nanotube	Graphene	Nanocone
carbon atoms	760	416	1077
hydrogen atoms	40	58	57
edge, α, or base radius, r (nm)	0.68	3	2.09
height, h (nm)	4.5	3	3.55
geometrical surface, *S* (nm${}^{2}$)	$S_{tube}=2\pi r^{\prime }h$	$S_{sheet}=2\alpha h$	$S_{cone}=\pi r^{\prime }\left(r^{\prime }+\sqrt{h^{2}+r^{\prime 2}}\right)$
$S^{out}$($r^{\prime }=r+\sigma_{C}$)	23.98	9.0	46.16
$S^{in}$($r^{\prime }=r-\sigma_{C}$)	14.36	-	36.13

**Table 2 molecules-27-00956-t002:** Box sizes after equilibration and the number of solvent molecules contained in the box with the carbon nanotube, the graphene, and the carbon nanocone.

Structure	Nanotube	Graphene	Nanocone
Volume X × Y × Z (nm${}^{3}$)	9.01 × 9.01 × 22.53	9.09 × 9.09 × 22.73	8.91 × 8.91 × 22.27
cycloexane molecules	4293	4338	4235
water molecules (spc)	34,235	35,410	32,702

**Table 3 molecules-27-00956-t003:** Computing performance of the umbrella simulations (tasks) for the carbon nanotube, the graphene, and the nanocone, using 20 computer processor units (cpus) per task.

Structure	Nanotube	Graphene	Nanocone
ns/day	5.278	5.183	5.433
hours/ns	4.547	4.631	4.417

## Data Availability

Data can be available in Gromacs filetypes such as gro, xvg, xtc, dat, and txt. Also codes are available in .sh, .py and pl.

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
