# Peer review of "Umbrella Sampling Simulations of Carbon Nanoparticles Crossing Immiscible Solvents"

_molecules, 2022, doi:10.3390/molecules27030956_

Round 1

Reviewer 1 Report

This work performs the umbrella-sampling molecular dynamics (MD) simulation to investigate the change in the solvation free energy of a carbon nanoparticle when the nanoparticle is moved from a hydrophobic solvent (cyclohexane) to a hydrophilic solvent (water). Comparing the results for a carbon nanocone, a carbon nanotube, and graphene, the authors discuss the effect of specific nanoparticle shapes and the solvation structure.

The simulation results seem reasonable, but I am not satisfied with some of the physical interpretations of the results. Please consider the following issues.

1) The authors proposed the water accessible surface area (WASA) as a key variable that affects the solvation free energy of a carbon nanoparticle. It seems that the nanoparticle-solvent interaction potential energy can play a similar role to WASA and is even more quantitative. In general, many aspects of solvation effects would be most easily explained by the competition between solvent-solvent, solvent-solute, and solute-solute (this is not relevant here) interactions. What is the advantage of WASA compared to the water-nanoparticle interaction energy?

2) The authors found that a carbon nanoparticle rotates as it moves from cyclohexane to water solvents. The authors explained “they rotate in an appropriate direction to reduce the amount of hydrophobic interactions at their front, making the force to decrease.” In my opinion, however, this explanation does not capture the true physics. Rather, the nanoparticle rotates so that the interaction with cyclohexane, which is more favorable than the water-nanoparticle interaction, is maximized at each z position.

3) Lines 269-271: “As the structures move, they agitate the ambient solvent. Due to the agitations the entropy increases and contributes to the system energy.” The meaning of this description is unclear to me, because the agitation is a non-equilibrium process whereas the entropy that constitutes the solvation free energy is an equilibrium property. More explanation would be necessary.

4) Lines 274-276: “Due to the presence of cyclohexane molecules inside the confinement, we encounter additional cyclohexane - water interactions which increase the potential energy.” The cyclohexane molecules remain absorbed on the nanoparticle surface also increase the graphene-cyclohexane interaction and decrease the graphene-water interaction. The authors ignore the potential energy change associated with these interactions. 

5) In Fig. 2, the pulling force at each z position in the pulling simulation is plotted. The integration of this force with respect to z coordinate would approximately correspond to a PMF, wouldn’t it? Can the authors compare this PMF with the PMF shown in Fig. 7? This comparison might support the sentence in Lines 226-228 “Based on this outcome, we infer that the initial configurations, created by the pulling simulation, are indeed, the most frequently sampled configurations in the umbrella simulations.”. 

6) Information about the simulation timestep and the methodology of calculating Coulombic interactions are missing. 

7) At some stages of the MD simulation (Lines 139-142), the system pressure was equilibrated by attaching different Berendsen barostats for the nanoparicle, water, and silica. This means that each barostat controls the pressure of a spatially localized group of atoms, but the original version of GROMACS does not assume such a calculation of local pressure.

Reviewer 2 Report

This study presents results obtained with molecular dynamics simulations to compute the free energy of carbon nanoparticles crossing a 
hydrophobic - hydrophilic interface. The author accumulate a series of umbrella sampling simulations along the charted path of the nanoparticle and compute the solvation energy with respect to the two solvents. Although design and results obtained with the simulation are valuable the lack of proper discussion in terms of scientific literature and potential application makes this work less valuable. It cannot be published in its current form.

The discussion section is very general and doesn't refer to any scientific sources.

The author writes "Due to the abundance of experimental data for solvents interacting with carbon surfaces, and because carbon surfaces are relatively simple, many of the pioneering simulations of surface phenomena 
involved studies of solvents interacting with homogeneous and heterogeneous carbon surfaces." - Can you add some references to back this claim?

In general Discussion section doesn't add anything valuable to the paper and has to be completely rewritten. Alternatively, it can be incorporated into the Results section where it is confronted with the scientific literature on this subject. Although is not a Conclusion section, it reads as such.

Round 2

Reviewer 2 Report

The author improved the paper as suggested. However, the discussion section is a bit short. Therefore it can be incorporated into the results section.

Ref 56 doesn't contain any literature details.

I recommend publication after minor revison.
